# Evaluating the Stability and Digestibility of Long-Chain Omega-3 Algal Oil Nanoemulsions Prepared with Lecithin and Tween 40 Emulsifiers Using an In Vitro Digestion Model

**DOI:** 10.3390/foods13152407

**Published:** 2024-07-29

**Authors:** Qiqian Zhou, Katie E. Lane, Weili Li

**Affiliations:** 1Functional Food Research Centre, Department of Clinical Sciences and Nutrition, University of Chester, Chester CH1 4BJ, UK; qiqian.zhou@163.com; 2Research Institute for Sport and Exercise Sciences, Liverpool John Moores University, Student Life Building, Copperas Hill, Liverpool L3 5AH, UK; k.e.lane@ljmu.ac.uk

**Keywords:** LCn-3PUFA, algal oil, nanoemulsion, stability, digestibility

## Abstract

The health benefits of long-chain omega-3 polyunsaturated fatty acid (LCn-3PUFA) intake have been well documented. However, currently, the consumption of oily fish (the richest dietary source of LCn-3PUFA) in the UK is far below the recommended level, and the low digestibility of LCn-3PUFA bulk oil-based supplements from triglyceride-based sources significantly impacts their bioavailability. LCn-3PUFA-rich microalgal oil offers a potential alternative for populations who do not consume oily fish, and nanoemulsions have the potential to increase LCn-3PUFA digestibility and bioavailability. The aims of this study were to produce stable algal oil-in-water nanoemulsions with ultrasonic technology to increase DHA digestibility, measured using an in vitro digestion model. A nanoemulsion of LCn-3PUFA algal oil was developed with 6% *w*/*w* emulsifiers: lecithin (LE) or an equal ratio of Tween 40 (3%) and lecithin (LTN) (3%), 50% *w*/*w*, algal oil and 44% *w*/*w* water using rotor–stator and ultrasound homogenization. The in vitro digestion experiments were conducted with a gastric and duodenal digestion model. The results showed the creation of nanoemulsions of LCn-3PUFA algal oils offers potentially significant increases in the bioavailability of DHA in the human body. The increase in digestibility can be attributed to the smaller particle size of the nanoemulsions, which allows for higher absorption in the digestive system. This showed that the creation of nanoemulsions of LCn-3PUFA algal oils offers a potentially significant increase in the bioavailability of DHA in the human body. The LE and LTN nanoemulsions had average droplet sizes of 0.340 ± 0.00 µm and 0.267 ± 0.00 µm, respectively, but the algal oil mix (sample created with same the components as the LTN nanoemulsion, hand mixed, not processed by rotor–stator and ultrasound homogenization) had an average droplet size of 73.6 ± 6.98 µm. The LTN algal oil nanoemulsion was stable in the gastric and duodenal phases without detectable destabilization; however, the LE nanoemulsion showed signs of oil phase separation in the gastric phase. Under the same conditions, the amount of DHA digested from the LTN nanoemulsion was 47.34 ± 3.14 mg/g, compared to 16.53 ± 0.45 mg/g from the algal oil mix, showing DHA digestibility from the LTN nanoemulsion was 2.86 times higher. The findings of this study contribute to the insight of in vitro DHA digestion under different conditions. The stability of the LTN nanoemulsion throughout digestion suggests it could be a promising delivery system for LCn-3PUFAs, such as DHA, in various food and pharmaceutical applications.

## 1. Introduction

In the search for optimal health and wellbeing, long-chain omega-3 polyunsaturated fatty acids (LCn-3PUFA), specifically docosahexaenoic acid (DHA), mainly found in marine sources, have been extensively studied for their health benefits [1,2,3,4,5,6]. Traditional sources, such as fish and fish oil, are now being reconsidered due to sustainability concerns and only a quarter (25.2%) of the population are oily fish consumers in the UK [7,8,9]. Algal oil, derived from microalgae, provides an excellent plant-based alternative to marine oils for non-fish consumers and those who follow a vegan or vegetarian diet [10]. Algal oil also provides a sustainable and ethical option for obtaining LCn-3PUFA [9]. The European Food Safety Authority (EFSA) concluded that 250 mg/day of LCn-3PUFA is required for the maintenance of cardiac heart function and normal brain function and development, and 2 g/day of docosahexaenoic acid (DHA) is needed for the maintenance of normal (fasting) blood concentrations of triglycerides in adult men and women [11,12,13].

Previous studies show that existing LCn-3PUFA encapsuled bulk oil supplement products on the market may lack efficacy in terms of dosage [14,15]. Additionally, their digestibility has been observed to vary across different delivery formats in both in vitro testing and human trials [16,17,18,19]. As a result, there is growing interest in exploring plant-based algal oil alternatives that offer high digestibility and efficacy, highlighting the need for further research and the development of algal-sourced plant-based alternatives that can provide a reliable and effective source of LCn-3PUFA.

Nanoemulsions, characterized by extremely small droplet sizes, give a promising avenue for enhancing the stability and bioavailability of lipophilic functional components and are being increasingly studied to protect and deliver LCn-3PUFA [18,19,20,21,22,23,24,25,26,27]. High-energy approaches such as ultrasound processing offer one of the most versatile means of producing food-grade nanoemulsions because they can be used with a wide variety of different oil and emulsifier types [28]. Emulsifiers are surface-active compounds that play an important role in the formation and stabilization of nanoemulsions [29]. Nonionic surfactants such as synthetic Tween 40 have been shown to have better stability under acidic conditions than natural surfactants [29]. The authors have previously established a method to produce nanoemulsions of LCn-3PUFA algal oil with optimized processing conditions and selected emulsifiers (lecithin, Tween 40, with equal ratios of both) to increase the oil load to 50% (*w*/*w*) using ultrasonic processing [24]. Furthermore, the authors have also shown that DHA absorption from algal oil nanoemulsions created using lecithin and ultrasound processing was significantly higher than with bulk algal oil, in an in vivo single-blind, randomized crossover human trial [30]. In addition, the results of in vitro digestion studies with algal oil and lecithin nanoemulsions prepared with microfluidization demonstrated that the stability of the nanoemulsion had a positive impact on the indicator of bioavailability [19]. However, at low pH (pH 1.6) the destabilization of the nanoemulsion resulted in a decrease in DHA hydrolysis compared with the nanoemulsion at pH 4. Studies have yet to investigate combinations of natural and synthetic emulsifiers to stabilize LCn-3PUFA nanoemulsion systems created using ultrasound processing at low pH conditions in digestion.

The current study aimed to conduct a novel evaluation of the stability and in vitro digestibility of LCn-3PUFA algal oil-in-water nanoemulsions created using rotor–stator and ultrasound homogenization with natural and synthetic emulsifiers in comparison to an unprocessed hand-mixed algal oil sample containing the same components (algal oil mix).

The specific aims of our study were as follows: Firstly, to prepare and create a hand-mixed sample using the natural emulsifier lecithin, synthetic emulsifier Tween 40 (equal ratios) and algal oil mix sample. Secondly, to further process samples by using rotor–stator and ultrasound homogenization [31] to create nanoemulsions using emulsifiers, specifically lecithin, and lecithin and Tween 40, in equal ratios. Thirdly, to assess the stability of the nanoemulsions under various environmental digestion conditions. Finally, to employ an in vitro digestion model to simulate and evaluate the digestion of the algal oil nanoemulsions and quantify the release of LCn-3PUFA in the algal oil nanoemulsions compared to the algal oil mix. This study expects to show that processing algal oil, lecithin and Tween 40 using rotor–stator and ultrasound homogenization will significantly increase DHA digestibility and bioavailability in comparison to the algal oil mix sample.

## 2. Materials and Methods

### 2.1. Materials

Algal oil (Life DHATM S35-O300) was purchased from DSM Ltd. (Columbia, SC, USA). L-α-phosphatidylcholine (P3644-100G) lecithin from soybean and type IV-S, ≥30% (enzymatic), polyoxyethylene sorbitan monopalmitate (Tween 40, P1504) were purchased from Sigma-Aldrich, Gillingham, UK. Sodium chloride (99.5%) was purchased from ACROS Spain (Barcelona, Spain). Hexane (HPLC Grade) was purchased from Fisher Scientific (Loughborough, UK). Methanol (HPLC Grade), sulphuric acid 95% and sodium sulphate anhydrous were purchased from VER BDH PROLABO chemicals (Lutterworth, UK).

### 2.2. Methods

#### 2.2.1. Preparation of Lecithin (LE), and Combined LE/Tween 40 (LTN) Algal Oil Nanoemulsions and Algal Oil Mix

Samples were prepared according to methods previously developed by the authors [24,31], using emulsifiers lecithin and lecithin and Tween 40 combined. Nanoemulsions consisted of 6% (*w*/*w*) lecithin (LE) or combined emulsifiers with equal ratios of Tween 40 (3%) and lecithin (3%) (LTN), combined with 50% (*w*/*w*) algal oil and 44% (*w*/*w*) distilled water. An emulsion pre-mix was created using lecithin and algal oil in 30:70 mass ratios, respectively, hand mixed, then placed in a shaking water bath at 56 °C for 2 h with a shaking speed of 30 rpm. For the algal oil mix and LTN samples, 430 g algal oil and 440 g distilled water combined with 30 g Tween 40 were added to 100 g of the pre-mix. For the LE sample, 360 g algal oil and 440 g distilled water were added to 200 g of the pre-mix. Both samples were hand stirred and placed in the same water bath for 2 h. Each sample was hand stirred for 30 s every hour. A quantity of the combined Tween 40 and lecithin sample, which was not fully emulsified and did not appear to be phase stable, was then removed from the water bath (algal oil mix). The remaining samples were then homogenized for two minutes at a maximum speed (1200 rpm) in a high-shear rotor–stator homogenizer (Silverson Machines Ltd., Chesham, UK), then processed using a BSP-1200 ultrasonic processor (Industrial Sonomechanics Ltd., New York, NY, USA) at 19,650 Hz, 100% energy, for 10 min at <50 °C controlled by a water-cooling system to create nanoemulsions (LE and LTN).

#### 2.2.2. In Vitro Digestion Model

Simulated digestion fluids were prepared as described in Table 1 and in accordance with Lin et al. [19] and Yang et al. [32]. Nanoemulsion, algal oil mix and control distilled water samples of 1 g were each made up to a 5 mL meal sample with 4 mL distilled water and kept at 37 °C in a water bath for 15 min, then mixed with 7.5 mL simulated gastric fluids (SGF), containing 40 mg of dispersed pepsin and 157 mg of pyrogallol (as an antioxidant). The pH of the digestion mixture was lowered to 1.6 with 0.1 M HCL solution, which represents the typical state of an empty stomach [33], and the gastric digestion of the mixture was started in a shaking water bath at 37 °C and 200 rpm. Following the gastric digestion phase, 3.5 mL of simulated bile fluid (SBF) with phospholipids and 7.5 mL simulated duodenal fluid (SDF) with pancreatin and pancreatic lipase was added to digestion mixture to achieve 10 mg/mL bile extract, 3.8 mg/mL phospholipids, 5 mg/mL pancreatin and 6 mg/mL pancreatic lipase. The pH of this mixture was adjusted to 6.85 with the addition of 1 N NaOH and the digestion mixture in the duodenal phase was kept at 37 °C for 3 h. After digestion, all the digested samples were kept at 4 °C in a refrigerator before centrifuging.

#### 2.2.3. Measurement of Droplet Size

The droplet size of the algal oil mix and nanoemulsion samples before, during and after digestion were determined by a Mastersizer 3000 laser light-scattering analyzer (Malvern Instrument Ltd., Malvern, UK) using methods by Lane et al. [25]. Samples were measured using a small sample dispersion unit at 2400 rpm, with an absorption parameter value of 0.001 and refractive index ratio of 1.488 for algal oil. Droplet size was reported as *d*_32_, the volume/surface diameter mean (Sauter mean), and *d*_43_, the mean diameter of the particle size, based on volume-weighted mean results (De Brouckere mean), to measure the specific surface area in line with previous studies [26,31,34].

#### 2.2.4. Isolation of the Aqueous Phase from Digested Fluid

Samples (23.5 mL per sample) were transferred into centrifuge tubes and balanced. The samples were centrifuged at 144,000× *g* at 7 °C for 1 h using an ultracentrifuge SW 32Ti rotor (Beckman Coulter, Brea, CA, USA) to separate the undigested oil from the aqueous phase. After centrifugation, the upper oil phase was carefully collected using a pipette. The volume and weight of the aqueous phase were measured with a serological pipette and all samples were stored in closed containers under −20 °C until further analysis.

#### 2.2.5. Extraction of Lipid of Aqueous Phase

Lipid extraction was performed using the same method as Lin et al. [19] with slight modifications as described. Firstly, 3 mL hexane–methanol (1:2) was added to 1 mL of the aqueous phase. The mixture was vortexed for 60 s, 1 mL hexane was added and samples were vortexed for 15 s. Phase separation was facilitated by adding 1 mL 0.5% NaCl followed by 30 s of vortexing. Finally, samples were centrifuged at 2000× *g* for 6 min, and aliquots from the hexane phase were withdrawn and kept at −20 °C until analysis.

#### 2.2.6. Determination of Fatty Acid Composition by GC

Lipid extraction was performed using the derivatization of fatty acid for FAME analysis by using the AOAC Official Methods of Analysis [35], which was developed at the NoWFood Research Centre, University of Chester, with slight modifications. A 1.0 g sample of hexane extract was added to 10 mL reagent A (2.5% *w*/*v* KOH solution in methanol). A cap was added and tightened; the tubes were placed in Kevlar sleeves then into a Mars 6 microwave (CEM Ltd., Edgware, UK). The temperature was increased to 90 °C over 5 min and held for 10 min. After cooling to room temperature, 15 mL reagent B (2% sulphuric acid *v*/*v* in methanol) was added. The sample tubes were resealed and placed in the microwave. The temperature was increased to 120 °C and held for 6 min. After cooling to room temperature, 10 mL hexane was added, and the tube inverted once. Sufficient saturated salt solution was added to bring the hexane layer to the top of the tube. The upper hexane layer containing the fatty acid methyl esters was separated for GC analysis.

#### 2.2.7. GC-FID Analysis

The samples were analyzed using a GC Clarus 480 system (PerkinElmer Inc., Shelton, CT, USA), equipped with an auto sampler, flame ionization detector (FID) and a 30 m, 0.25 mm id 0.25 µm film thickness GC capillary column (SGE Analytical Science Pty Ltd., Ringwood, VIC, Australia) and TotalChrom Navigator software system (Version 6.3.2, PerkinElmer Inc., Shelton, CT, USA). The injector and detector temperatures were 220 °C and 250 °C, respectively, with a 1.5 µL injection for each instance, and the hydrogen flow pressure was set at 8.4 psi. The column temperature was programmed to increase from 60 °C to 170 °C at a rate of 20 °C/min and to 200 °C at a rate 1 °C/min, and held at 200 °C for 1 min; the total run time was 36.5 min. The fatty acids were identified by reference to the retention time of standards (Supelco 37 Component FAME Mix, Sigma, Gillingham, UK). Analysis was performed in triplicate on individual vials for each time point.

#### 2.2.8. Statistical Analysis

All measurements were completed in triplicate. Results are expressed as mean ± standard deviation. A one-way ANOVA test was conducted (IBM, SPSS statistics, v 24, New York, NY, USA), then the Tukey post hoc test was used to compare sample means to compare and check for significant differences in the distribution of data and for droplet size testing, with *p* values of <0.05 considered significant.

## 3. Results

### 3.1. Droplet Size and Stability Measurements of Algal Oil Mix and Nanoemulsions

The droplet size distribution and span data of the control, algal oil Mix, and LE and LTN nanoemulsions were measured by using a Mastersizer 3000 laser light-scattering analyzer and the results are shown in Table 2. Significant differences (*p* < 0.05) were found in the mean droplet size between samples of algal oil mix and LE and LTN algal oil nanoemulsions produced with high shear mixing and ultrasonic processing. The appearance of the LE and LTN algal oil nanoemulsions during digestion with the pH 1.6 gastric phase and pH 6.8 duodenal phase digestion model can be found in Figure 1A and 1B, respectively. Oil phase separation was observed during the digestion of the LE algal oil nanoemulsion at pH 1.6 in the gastric phase, and it did not disappear after 60 min in the pH 6.8 duodenal phase (Figure 1A). In contrast, no oil phase separation was detected with the LTN nanoemulsion during digestion in both the pH 1.6 gastric phase and the pH 6.8 duodenal phase (Figure 1B), demonstrating that the LTN nanoemulsion remained stable during digestion. These results indicate that the LTN nanoemulsion may have potential as a more stable and efficient delivery system for bioactive compounds in the gastrointestinal tract. Due to the phase separation of the LE sample in the gastric phase, the remaining analyses and results are shown for the LTN and algal oil mix samples.

### 3.2. Droplet Size Distribution of the LTN Nanoemulsion and Algal Oil Mix during In Vitro Digestion

The changes in droplet sizes of the LTN nanoemulsion and algal oil mix during digestion are shown in Table 3. The average droplet size of LTN nanoemulsions before digestion was 0.267 ± 0.00 µm and it was measured at 0.245 ± 0.00 µm and 0.238 ± 0.01 µm at 5 min and 60 min in the gastric phase (pH 1.6). There were no significant changes in the droplet sizes, showing the LTN sample had good stability in the gastric digestion phase. Contrastingly, the algal oil mix had an average droplet size of 73.6 ± 6.98 µm before digestion, but it was significantly reduced to 50.1 ± 1.81 µm and then 28.83 ± 0.06 µm at 5 min and 60 min digestion, respectively (*p* < 0.05), showing the algal oil in the mix sample was further emulsified during gastric digestion with shaking. However, the droplet size was still significantly higher (*p* < 0.05) and over 100 times larger than the average droplet size of the LTN nanoemulsion. Following gastric digestion, 3.5 mL of SBF with phospholipids, bile salts and 7.5 mL SDF with pancreatin and pancreatic lipase were added to the digestion mixture with SG; the average droplet size of the digestion mixture for the LTN nanoemulsion increased to 0.326 ± 0.01 µm at 5 min, then to 14.20 ± 0.26 at 60 min and then decreased to 8.87 ± 2.56 µm after 180 min of the duodenal phase (pH 6.8), showing the mean droplet size changed significantly (*p* < 0.05). It appears that the addition of bile extract, phospholipids, pancreatin and pancreatic lipase contributed to the increase in droplet sizes from 0.326 ± 0.01 µm to 14.20 ± 0.26. Meanwhile, the droplet size of the digestion mixture of the algal oil mix was in a larger range with a significant (*p* < 0.05) decrease from 27.30 ± 0.30 µm to 18.97 ± 8.19 µm at the end of the duodenal phase.

Figure 2 shows the droplet size distribution of the LTN nanoemulsion during in vitro digestion using fluids (SGF, SBF and SDF) without adding pepsin, pyrogallol, bile extract, phospholipids, pancreatic lipase and pancreatin. The results showed that the droplet size of the digestion mixture with the LTN nanoemulsion distributed was in the range of 0.1–1 µm and remained stable during digestion in both gastric and duodenal phases during in vitro digestion, which means the pH variation from 1.6 for the gastric phase to 6.8 for the duodenal phase did not cause any changes in nanoemulsion droplet size. The increase in the average droplet size of the digestion mixture with the LTN nanoemulsion and digestion fluid in the duodenal phase (Table 3) may be attributed to the addition of digestion enzymes.

In contrast, Figure 3 shows the droplet distribution of the distilled water control sample with digestion fluids. There is no obvious peak for droplets in the range of 0.1 μm, but large peaks in the range of 1–1000 μm were observed at the beginning of duodenal phase. However, at 120 min of the duodenal phase, a peak at 0.1 µm for droplets appeared and the large peaks in the range above 1 μm still existed. Therefore, the digestive fluid containing phospholipids can facilitate the formation of a small volume of droplets for a droplet size of 0.1 μm by self-emulsifying [29,33].

Figure 4 shows that the droplet size distribution of the LTN nanoemulsion with added pepsin, pyrogallol, bile extract, phospholipids, pancreatic lipase and pancreatin was stable for 60 min in the gastric phase, but it changed significantly (*p* < 0.05) in the duodenal phase during in vitro digestion as shown in Table 3. The peak droplet size of the LTN nanoemulsion for the range of 0.1–1 µm decreased quickly as digestion in the duodenal phase started and it disappeared after 60 min in the duodenal phase; meanwhile, droplets with average sizes of 0.15 µm appeared 120 min into the duodenal phase and subsequently the peak for the droplets with a smaller average size of 0.12 µm occurred at 180 min into the duodenal phase, showing fatty acids and monoglycerides were gradually released from the triglycerides of the LTN nanoemulsion and formed droplets with an average size of 0.12 µm.

Figure 5 shows the droplet size distribution of the algal oil mix sample with added pepsin, pyrogallol, bile extract, phospholipids, pancreatic lipase and pancreatin changed during in vitro digestion. In the gastric phase, the peak was for droplet sizes in the range of 10–290 µm and it moved to a smaller range of 5–170 µm; however, in the duodenal phase, only a small peak in the range of 0.08–0.62 μm appeared at the end of digestion and the peaks in the range of 1–1000 µm still existed, showing that a much smaller portion of fatty acids and monoglycerides were released from the triglycerides of algal oil mix compared to those from LTN nanoemulsion.

### 3.3. The Droplet Size Distribution of the Aqueous Phase of LTN Nanoemulsion and Algal Oil Mix Samples after Digestion

After digestion, the samples were subjected to ultracentrifugation to separate the nonhydrolyzed triglycerides which were present in the top oil phase and the hydrolyzed fatty acids which were in the aqueous phase (Figure 6). The droplet size distribution of the aqueous phase of LTN nanoemulsion and algal oil mix samples are presented in Figure 7. For the aqueous phase from the digested LTN nanoemulsion, one large peak in the range of 0.02 to 0.49 µm and with a mean droplet size of 0.10 ± 0.002 µm was observed, which is in the same range of droplet sizes as in Figure 4. In contrast, for the aqueous phase from the digested algal oil mix sample, there were two distinct peaks, one in the range of 0.04 to 0.70 µm with a mean droplet size droplet size of 0.15 ± 0.002, with a similar droplet size range to that in Figure 5, and the other in the range of 10–100 µm.

### 3.4. Digestibility of DHA of LTN Nanoemulsion and Algal Oil Mix during In Vitro Digestion

The fatty acid composition of samples was analyzed by gas chromatography, in which a standard mixture of known FAMEs [35] was run alongside the samples. The specific retention time of the peaks was used to identify the fatty acids. Figure 8 shows that the gas chromatogram for the LTN nanoemulsion had six peaks and they were identified to be myristic acid (14:00); palmitic acid (16:00); oleic acid (18:1n−9); linoleic acid (18:2n−6); docosapentaenoic acid (DPA, 22:5n−6) and DHA (22:6n−3) according to their retention time.

The percentage of each fatty acid was calculated by using the peak area divided by the total peak area of six main peaks. The results are shown in Table 4. The proportion of DHA in the LTN nanoemulsion and algal oil mix was 39.40 ± 4.15 before digestion, which is significantly (*p* < 0.05) higher than that in the aqueous phase of the digested LTN nanoemulsion (28.89 ± 0.74%) and algal oil mix (19.95 ± 0.40%). It is interesting to note that the proportion of DHA from the aqueous phase of the LTN nanoemulsion (28.89 ± 0.74%) is significantly higher (*p* < 0.05) than that of the aqueous phase of digested algal oil mix (19.95 ± 0.40), indicating the DHA digestibility of LTN nanoemulsion was improved compared to that of the algal oil mix.

Furthermore, it was observed that the proportions of oleic acid (18:1n−9) and linoleic acid (18:2n−6) were significantly higher (*p* < 0.05) in the aqueous phase of the digested LTN nanoemulsion and algal oil mix compared to in the LTN nanoemulsion and algal oil mix before digestion, indicating that the molecular structures of oleic acid (18:1n−9) and linoleic acid (18:2n−6) are likely favourable to hydrolysis in the duodenal lipase for both the LTN nanoemulsion and algal oil mix, which is different from DHA digestion.

In addition, the DHA content of the aqueous phase of the digested LTN nanoemulsion and algal oil mix was quantified using gas chromatography with the external DHA standard curve. The DHA content of the aqueous phase of the digested LTN nanoemulsion and algal oil mix are shown in Table 5. The DHA content of the LTN nanoemulsion and algal oil mix before digestion was 162.12 ± 17.07 mg/g; however, for the algal oil mix sample, only 16.53 ± 0.45 mg/g out of 162.12 ± 17.07 mg/g was digested and transferred into the aqueous phase, indicating only 10% DHA digestibility. Interestingly, 47.34 ± 3.14 mg DHA was digested and transferred into aqueous phase from the LTN nanoemulsion, which further confirmed its DHA digestibility increased by around three times.

## 4. Discussion

This study aimed to evaluate potential increases in DHA digestibility by producing stable algal oil-in-water nanoemulsions using continuous ultrasonic processing and an in vitro digestion model. The algal oil nanoemulsions were prepared by following our previous methods [20] with a single emulsifier (6% LE) and combined emulsifiers (LTN, 3% Lecithin and 3% Tween 40). The mean droplet sizes of the LE nanoemulsion and LTN nanoemulsions were consistent with those in our previous studies [19,24,30]. The stability and digestibility of both nanoemulsions were tested using an in vitro simulated digestion model, which has been extensively used to monitor the digestibility/bioaccessibility of food and drug components under simulated gastrointestinal conditions [18,19,32,37,38]. Typically, ingested triglycerides go through hydrolysis by gastric and pancreatic lipase, leading to the release of free fatty acids and monoglycerides, which interact with bile salts and phospholipids to form mixed micelles [39,40]. When the droplet sizes of emulsified lipids decrease, the digestion rate of lipids increases due to the increased surface area of the smaller droplets [40,41]. Figure 1A shows the LE nanoemulsion had oil separation at pH 1.6 in the gastric phase, which agrees with findings by Lin et al. [19]. Yang et al. [32] also showed that low pH caused an increase in the droplet sizes of a nanoemulsion prepared with soy lecithin and curcumin during digestion using an in vitro model, which confirms soy lecithin-only emulsions are not stable when exposed to acidic gastric conditions. In contrast, the LTN nanoemulsion prepared with equal ratios of lecithin and Tween 40 in the current study was shown to be stable without any oil separation under the same conditions (Figure 1B). Tween 40 is a nonionic molecule, which is not affected by the variation of environmental pH [42], therefore stabilizing the LTN nanoemulsion during digestion.

The droplet sizes recorded for the LTN nanoemulsion ranged from 0.01 to 1 μm (Figure 2), and remained stable under pH conditions of 1.6 for gastric phases and 6.8 for duodenal phases. During the duodenal phase of the in vitro model, the droplets of the LTN nanoemulsion disappeared and micelles of hydrolyzed fatty acids formed which were measured to be 0.10 ± 0.002 µm in aqueous phase after ultracentrifugation. The results demonstrated the combination of Tween 40 and lecithin to be beneficial to increase the digestibility of DHA in triglycerides.

Lin et al. [19] showed that the emulsification of algal oil with lecithin promoted the transfer of DHA into the aqueous phase as it was stable in the gastric phase, resulting in the higher bioavailability of DHA in a human trial [30]. Similarly, Yang et al. [32] demonstrated that curcumin and co-delivering essential oil emulsions stabilized with soy lecithin increased the stability and bioaccessibility of curcumin and essential oils. Coupled with our results, these previous findings show that emulsification does play a role in enhancing the digestibility of oil and the aqueous phase transfer during digestion if emulsions remain stable in digestion [42].

In this study, the droplet size changes in the LTN nanoemulsion and algal oil mix during in vitro digestion presented in Figure 4 and Figure 5 revealed their digestion characteristics. In the duodenal phase, as the peak of droplets for the LTN nanoemulsion decreased and disappeared, the peak for micelles appeared and shifted to the smaller droplet size range, which is an indicator for the release of hydrolyzed fatty acids, including DHA and monoglycerides, from the triglycerides. However, the droplet size peak of micelles for the algal oil mix sample was much smaller at the end of duodenal phase compared to those from the LTN of the nanoemulsion.

After digestion, the droplet size of micelles from the aqueous phase of the digested LTN nanoemulsion and the digested algal oil mix was significantly different (*p* < 0.05) (Figure 7). The aqueous phase of the digested LTN nanoemulsion had a large fraction of micelles in the range of 0.02 to 0.49 µm, which should be ready for absorption, while the droplet sizes in the aqueous phase of the digested algal oil mix were in separate ranges, located in the ranges of 0.04 to 0.70 µm and 10–100 µm, respectively. Large oil droplets cannot be absorbed into the blood stream without forming micelles of around 0.1 µm, and smaller micelles are absorbed more readily [33,43,44].

In addition, DHA transferred to the aqueous phase was quantified based on weight in this study. The results show that for the LTN nanoemulsion, 47.34 mg DHA was digested and transferred into the aqueous phase. Under the same conditions, only 16.53 mg DHA was digested and transferred into the aqueous phase from the algal oil mix, demonstrating that the DHA digestibility of the LTN nanoemulsion is 2.86 times higher than that of the algal oil mix. Our study findings show that the LTN nanoemulsion is likely to have high bioavailability in potential human trials and its incorporation into food and supplement matrices offers the possibility of reducing daily intake whilst still achieving the associated health benefits. To meet the requirement for an EFSA health claim for DHA to maintain normal (fasting) blood concentrations of triglycerides [11,12,13], a daily intake of 2000 mg of DHA, equivalent to 5 g (5 × 1 g/capsule) of 35% DHA algal oil, is required, representing a challenge in the daily diet. Our study findings show the incorporation of the LTN nanoemulsion offers the possibility of reducing the daily intake to 1.7 g/day nanoemulsion whilst still achieving the associated health benefits. Therefore, the present LTN nanoemulsion offers an efficient delivery system of LCn-3PUFAs and has the potential to reduce the daily intake required for EFSA health claims.

This study showed the LE-only nanoemulsion had oil separation in the gastric phase at pH 1.6; in contrast, the LTN nanoemulsion prepared with equal ratios of lecithin and Tween 40 was stable without any oil separation under the same conditions. The in vitro testing was completed and the algal oil mix sample was compared to the nanoemulsion samples to show how the nanoemulsion prepared by using rotor–stator and ultrasound homogenization can potentially improve DHA digestibility compared to the bulk oil supplements currently available on the market. However, this study did not evaluate the samples after rotor–stator homogenization, prior to ultrasound processing, which might be a limitation.

Lecithins are natural compounds that can be extracted from various sources including eggs, sunflower and soybean [29]. Synthetic non-ionic surfactants such as Tween 40 have been shown previously [29] and in this study to be efficient emulsifiers that produce nano-scale droplets with excellent stability. However, there is increasing concern about the adverse health and environmental hazards posed by the consumption or use of synthetic substances in the food chain [26]. With consumers seeking familiar, naturally derived, minimally processed ingredients, further research is warranted to examine the stability and digestibility of algal oil nanoemulsions created using natural alternatives to synthetic surfactants in combination with lecithin.

## 5. Conclusions

This study examined the stability and digestibility of algal oil nanoemulsions created using homogenization and ultrasonic technology compared to an algal oil mix (of the same composition not processed by rotor–stator and ultrasound homogenisations) using an in vitro digestion approach. After the LE and LTN nanoemulsions were prepared by using the ultrasonic processor, LTN nanoemulsion was shown to be stable without any noticeable destabilization at pH 1.6 for 60 min in the gastric phase and at pH 6.8 for 60 min in the duodenal phase environment, but LE appeared with oil phase separation under the same conditions. As the LTN nanoemulsion was digested in the duodenal phase of the in vitro model, the droplets disappeared and micelles of hydrolyzed fatty acids formed, which were measured to be 0.10 ± 0.002 µm in the aqueous phase after ultracentrifugation. The quantification of DHA for the aqueous phase from digested samples clearly demonstrated that the DHA digestibility of the LTN nanoemulsion was 2.86 times higher than that of the algal oil mix sample. The results of this study suggest that LTN nanoemulsions could have a high efficacy for the bioavailability of LCn-3PUFA, such as with DHA; however, further research is warranted to examine natural emulsifiers in place of synthetic surfactants.

## Figures and Tables

**Figure 1 foods-13-02407-f001:**
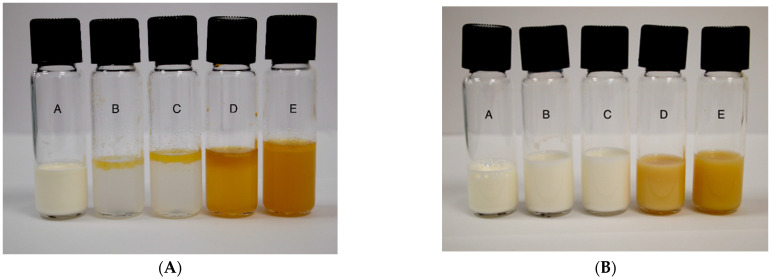
The appearance of LE nanoemulsion (**A**) and LTN (**B**) nanoemulsion during in vitro digestion. A. Before SGF addition, B. 5 min of pH 1.6 gastric phase, C. 60 min of pH 1.6 gastric phase, D. 5 min of pH 6.8 duodenal phase, E. 60 min of pH 6.8 duodenal phase.

**Figure 2 foods-13-02407-f002:**
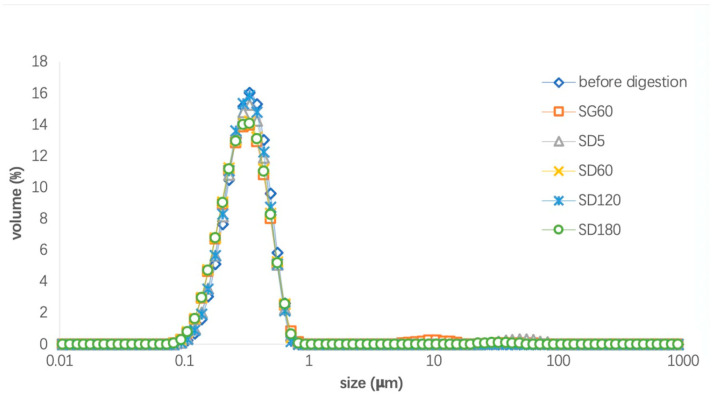
Droplet size distribution of nanoemulsion during digestion without the added digestion enzymes (pepsin, pyrogallol, bile extract, phospholipids, pancreatic lipase, pancreatin): SG60: 60 min of gastric digestion; SD5: 5 min of duodenal digestion; SD60: 60 min of duodenal digestion; SD120: 120 min of duodenal digestion; SD180: 180 min of duodenal digestion.

**Figure 3 foods-13-02407-f003:**
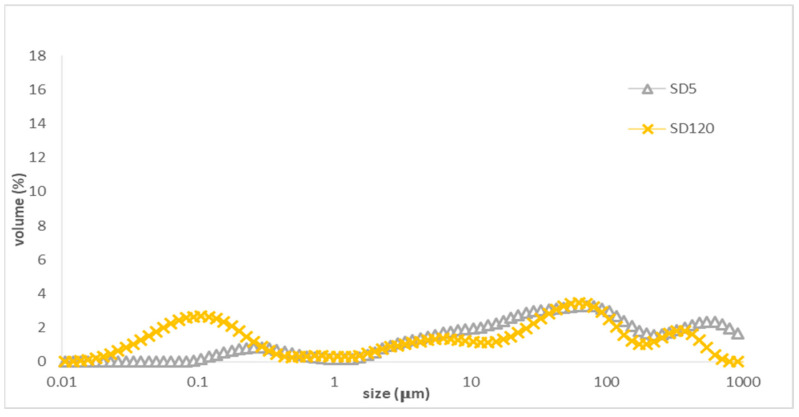
Droplet size distribution of control (distilled water) during in vitro digestion: SD5: 5 min of duodenal digestion; SD120: 120 min of duodenal digestion.

**Figure 4 foods-13-02407-f004:**
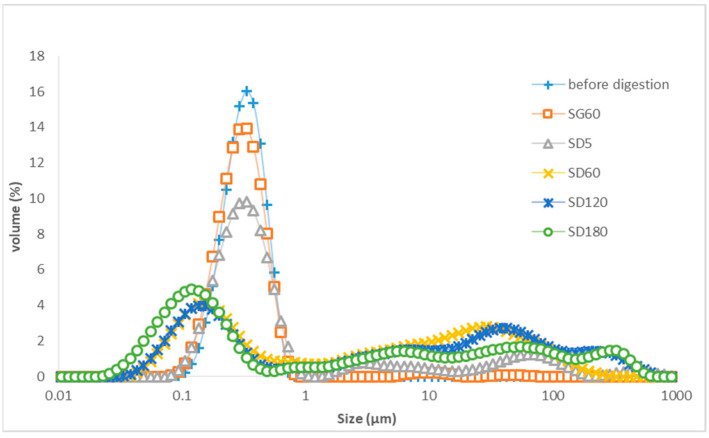
Droplet size of LTN nanoemulsion during in vitro digestion: SG60: 60 min of gastric digestion, SD5: 5 min of duodenal digestion; SD60: 60 min of duodenal digestion; SD120: 120 min of duodenal digestion; SD180: 180 min of duodenal digestion.

**Figure 5 foods-13-02407-f005:**
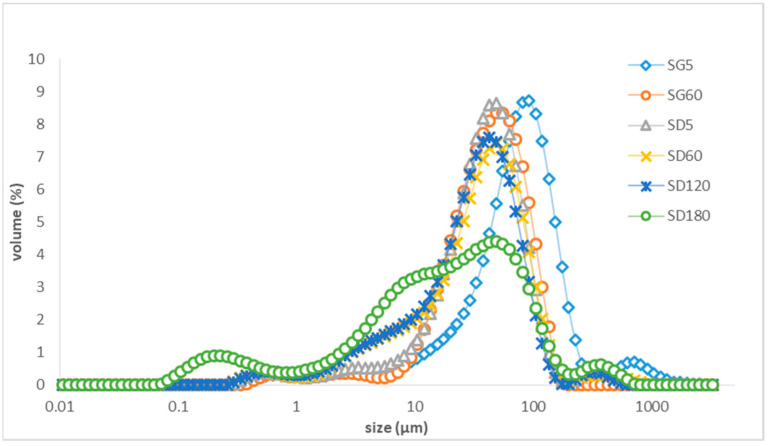
Droplet size of algal oil mix during in vitro digestion: SG60: 60 min of gastric digestion; SD5: 5 min of duodenal digestion; SD60: 60 min of duodenal digestion; SD120: 120 min of duodenal digestion; SD180: 180 min of duodenal digestion.

**Figure 6 foods-13-02407-f006:**
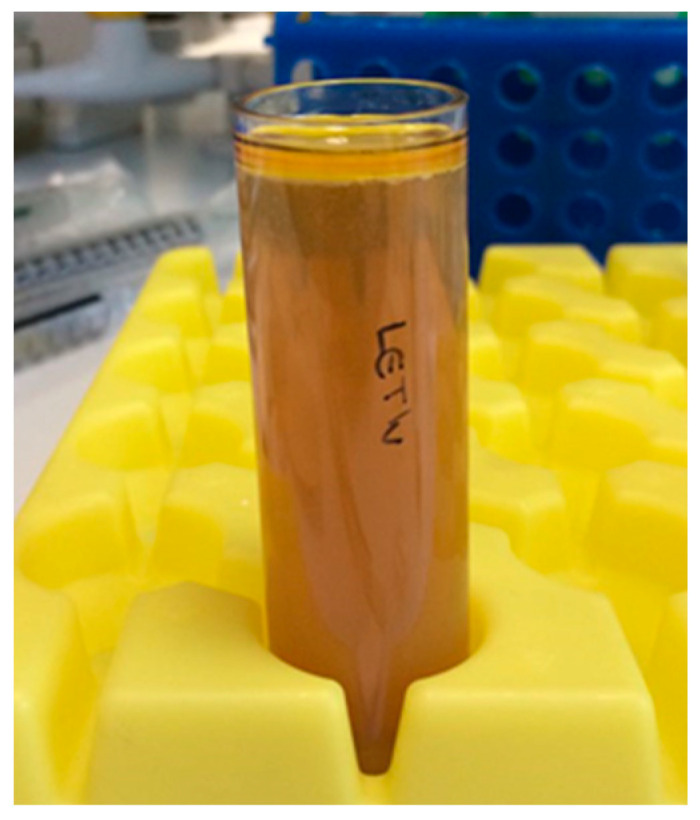
Separation of oil phase from water phase from digested LTN nanoemulsion after ultracentrifugation.

**Figure 7 foods-13-02407-f007:**
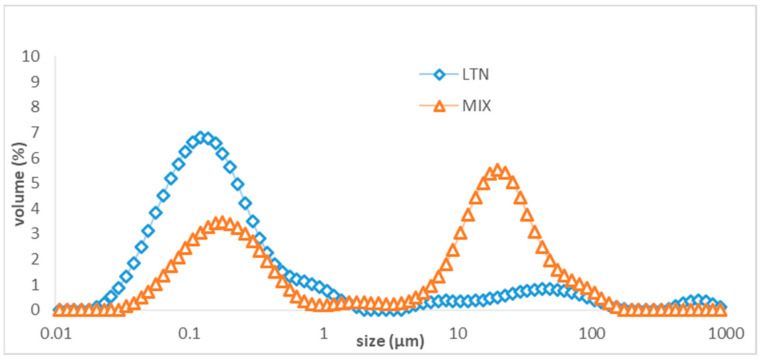
Droplet size of aqueous phase from digested LTN nanoemulsion and algal oil mix sample after ultracentrifugation.

**Figure 8 foods-13-02407-f008:**
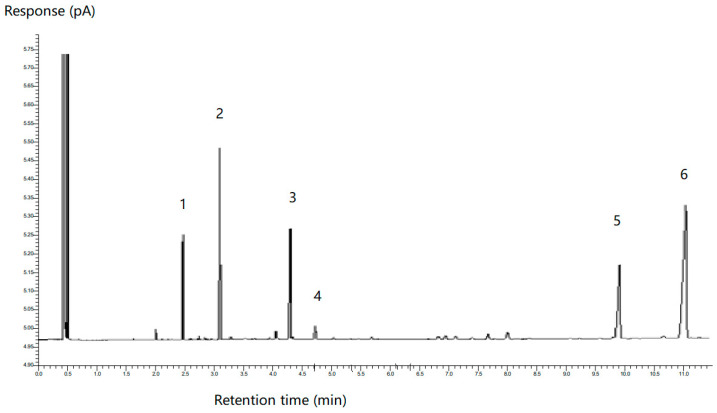
Peaks of fatty acids from LTN nanoemulsion: 1. C14:00 myristic acid; 2. C16:00 palmitic acid; 3. C18:1n−9 oleic acid; 4. C 18:2n−6 linoleic acid; 5. C22:5n−6 docosapentaenoic acid; 6. C22:6n−3 docosahexaenoic acid.

**Table 1 foods-13-02407-t001:** Composition and pH of the simulated gastric (SGF), duodenal (SDF) and bile (SBF) fluids used to the mimic fed state condition and concentration of constituents in the digestion mixture after the emulsion addition.

Component	SGF	SDF	SBF	Final Mixture
	(mM)	(mM)	(mM)	(mM)
NaCl	94.2	240	180	130.5
NaH_2_PO_4_	4.4	-	-	1.4
NaHCO_3_	-	80	137	45.7
KCl	22.1	15.1	10	13
KH_2_PO_4_	-	1.2	-	0.4
CaCl_2_·2H_2_O	5.4	2.7	3	3
NH_4_Cl	11.4	-	-	3.5
MgCl_2_	-	0.5	-	0.2
Total	137.6	340	330	197.5
pH	1.3	8.1	8.2	6.5

**Table 2 foods-13-02407-t002:** Mean of diameters of droplet size (*d*_32_) and span data [36] for algal oil mix, LE and LTN nanoemulsions before digestion.

	Algal Oil Mix	LE Nanoemulsion	LTN Nanoemulsion
Mean droplet size (*d*_32_ (µm))	73.6 ± 6.98	0.340 ± 0.00 ^a^	0.267 ± 0.00 ^a^
Span	1.24	2.91	1.78

^a^ Significantly different to the droplet sizes of the algal oil mix (*p* < 0.05).

**Table 3 foods-13-02407-t003:** Mean droplet (*d*_32_) of LTN nanoemulsion and algal oil mix samples during in vitro digestion with digestion enzymes.

	LTN Nanoemulsion	Algal Oil Mix
Digestion Stage	*d*_32_ (µm)	*d*_32_ (µm)
Before digestion	0.267 ± 0.00	73.6 ± 6.98
Gastric 5 min	0.245 ± 0.00	50.1 ± 1.81 ^c^
Gastric 60 min	0.238 ± 0.01	28.83 ± 0.06 ^cd^
Duodenal 5 min	0.326 ± 0.01	27.30 ± 0.30 ^cd^
Duodenal 60 min	14.20 ± 0.26 ^a^	21.67 ± 0.57 ^cd^
Duodenal 120 min	10.31 ± 0.52 ^ab^	19.82 ± 0.10 ^cd^
Duodenal 180 min	8.87 ± 2.56 ^ab^	18.97 ± 8.19 ^cd^

Measured data are means ± SD of triplicate testing. ^a^ Significant difference in droplet size of LTN (*p* < 0.05) compared with duodenal 5 min. ^b^ Significant difference in droplet size of LTN (*p* < 0.05) compared with duodenal 60 min. ^c^ Significant difference in droplet size of algal oil mix (*p* < 0.05) compared with before digestion. ^d^ Significant difference in droplet size of algal oil mix compared with 5 min digestion of gastric phase.

**Table 4 foods-13-02407-t004:** Fatty acid composition of LTN nanoemulsion/algal oil mix before digestion, aqueous phases of digested LTN nanoemulsion and algal oil mix.

Fatty Acid Sample	Myristic (%)	Palmitic (%)	Oleic (%)	Linoleic (%)	DPA (%)	DHA (%)
Before digestion	5.39 ± 0.68	17.12 ± 2.33	16.37 ± 2.14	4.98 ± 0.71	16.74 ± 1.89	39.40 ± 4.15
Aqueous phase of digested LTN	3.53 ± 0.04	11.15 ± 0.48	28.17 ± 0.57 ^a^	12.53 ± 0.32 ^a^	15.74 ± 0.58	28.89 ± 0.74 ^a^
Aqueous phase of digested algal oil mix	5.83 ± 0.20	19.06 ± 0.53	29.38 ± 0.62 ^a^	15.34 ± 0.36 ^ab^	10.46 ± 0.39 ^ab^	19.95 ± 0.40 ^ab^

^a^ Significant difference (*p* < 0.05) compared to before digestion; ^b^ significant difference (*p* < 0.05) compared with digested LTN. Measured data are the means ± SD of duplicate lipid extraction from duplicate digestions.

**Table 5 foods-13-02407-t005:** The DHA content of LTN nanoemulsion/algal oil mix before digestion and DHA content of the aqueous phase from digested LTN nanoemulsion and algal oil mix.

	Before Digestion DHA/LTN and Algal Oil Mix	Digested DHA/LTN	Digested DHA/ Algal Oil Mix
DHA (mg/g)	162.12 ± 17.07	47.34 ± 3.14 ^a^	16.53 ± 0.45 ^ab^

^a^ Significant difference (*p* < 0.05) in DHA of LTN or mix before and after digestion; ^b^ significant difference (*p* < 0.05) between digested DHA/LTN and digested DHA/algal oil mix.

## Data Availability

The data presented in this study are openly available in ChesterRep at https://chesterrep.openrepository.com/, (accessed on 15 June 2024).

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
