# Peer review of "Evaluating the Stability and Digestibility of Long-Chain Omega-3 Algal Oil Nanoemulsions Prepared with Lecithin and Tween 40 Emulsifiers Using an In Vitro Digestion Model"

_foods, 2024, doi:10.3390/foods13152407_

Round 1

Reviewer 1 Report (Previous Reviewer 2)

Comments and Suggestions for Authors

The revised study entitled ˮEvaluating the stability and digestibility of an omega-3 algal oil nanoemulsion using an in vitro digestion modelˮ, by Zhou Q, Lane KE, and Li W, explores the possibility of producing nanoemulsions of omega-3 algal oil using lecithin (LE) or lecithin and Tween 40 mixture (LTN) as emulsifiers. The nanoemulsions were obtained by heating in a water bath, mild stirring, homogenization with a rotor-stator homogenizer, and ultrasound processing. As a control, the Authors have used an unstable emulsion (algal oil MIX) produced by heating in a water bath and mild stirring, using LTN as an emulsifier. The prepared emulsions were subsequently used in stability and digestibility experiments. Although the Authors have improved the study, unfortunately, the main flaw has not been addressed. Furthermore, the revised Manuscript has some additional flaws.

1.      The control is inadequate and has not been changed. The control was an unstable coarse emulsion prepared only by heating in a water bath and mild stirring (lines: 103-113), whereas the investigated emulsions have been homogenized with a rotor-stator homogenizer, and ultrasound processing (lines 113-117). The Authors should have employed the rotor-stator homogenizer to obtain the control and make the study more valid.

2. The Abstract and Conclusions are misleading. The Authors imply in both, the Abstract and Conclusions, that the control, i.e., algal oil MIX, and the nanoemulsions differ only in ultrasound processing.  However, it is clear, based on the explanation in Methods, that the control was not processed by both, a rotor-stator homogenizer and ultrasound.

3.      It can be noticed based on the result that nanoemulsions having LE or LTN as emulsifiers were investigated. However, the Authors state, in the aims of the study, that the nanoemulsions were prepared by processing algal oil MIX emulsion (lines 81-84).

It is impossible to develop LE nanoemulsion if an emulsion was used for its development, which contains LTN, i.e., both lecithin and Tween 40.  

4.   The Authors have explained the development of the LTN nanoemulsion in the Methods, but not the development of the LE nanoemulsion.

5.      The revised Manuscript still has some typos. For example, the Authors use the term Ë®course emulsionË® (line: 111) instead of Ë®coarse emulsionË®.

Author Response

Dear Reviewers and Editor

We would like to thank you for providing a further opportunity to submit a revised draft of the manuscript ‘Evaluating the stability and digestibility of an omega-3 algal oil nanoemulsion using an in vitro digestion model.’ for publication in Foods.

We wish to thank the reviewers for their valuable feedback and have sought to incorporate their suggestions where possible, thereby improving the quality of our paper. We have made appropriate changes to the manuscript. Please see below, in red, our response to each of the reviewers’ comments accompanied by associated changes to the manuscript and feedback specific changes in line numbers added to refer to the revised manuscript file.

Reviewer 1

  1. The control is inadequate and has not been changed.The control was an unstable coarse emulsion prepared only by heating in a water bath and mild stirring (lines: 103-113), whereas the investigated emulsions have been homogenized with a rotor-stator homogenizer, and ultrasound processing (lines 113-117). The Authors should have employed the rotor-stator homogenizer to obtain the control and make the study more valid.

Thank you for this feedback, we hope we can clarify with our response and amendments to the draft. The aim of our study was to show how the digestibility and bioavailability of currently available bulk oil based LCn-3PUFA supplements (outlined in the introduction) could be improved with the provision of nanoemulsions, therefore the use of a homogenised coarse emulsion sample was not relevant. To achieve our study aims, we used bulk algal oil loosely mixed with the same components as the nanoemulsions without any processing. We completed novel in-vitro testing with the unprocessed bulk oil sample compared to the nanoemulsion samples show how homogenization and ultrasound processing can potentially improve DHA digestibility compared to bulk oil supplements currently available on the market. We have amended highlighted areas of the background detail and methods in the manuscript to clarify this.

  1. The Abstract and Conclusions are misleading. The Authors imply in both, the Abstract and Conclusions, that the control, i.e., algal oil MIX, and the nanoemulsions differ only in ultrasound processing.  However, it is clear, based on the explanation in Methods, that the control was not processed by both, a rotor-stator homogenizer and ultrasound.

Thank you for pointing this out. We have added further details to the abstract and conclusion to clarify.

(created with same the components as the LTN nanoemulsion, hand mixed, not processed by rotor-stator homogenizer and ultrasound)

(Please see lines: 26-27 and 511). 

  1. It can be noticed based on the result that nanoemulsions having LE or LTN as emulsifiers were investigated. However, the Authors state, in the aims of the study, that the nanoemulsions were prepared by processing algal oil MIX emulsion (lines 81-84).

We have added further detail to the study aims to clarify this

The specific aims of our study were, firstly, to prepare and create a hand mixed sample using natural emulsifier lecithin, synthetic emulsifier Tween 40 (equal ratios) and bulk algal oil (MIX sample). Secondly, to further process samples by homogeni-zation and ultrasonic continuous processing to create nanoemulsions using emulsifi-ers, lecithin and lecithin and Tween 40 in equal ratios. Thirdly to assess the stability of the nanoemulsions under various environmental digestion conditions. Finally, to employ an in vitro digestion model to simulate and evaluate the digestion of the algal oil nanoemulsions and quantifying the release of LCn-3PUFA in the algal oil nanoemulsions compared to the bulk algal oil MIX. This study expects to show that processing algal oil, lecithin and Tween 40 using homogenization and ultrasound technology will significantly increase DHA digestibility and bioavailability in com-parison to the bulk oil MIX sample.

(lines 84 to 94).

It is impossible to develop LE nanoemulsion if an emulsion was used for its development, which contains LTN, i.e., both lecithin and Tween 40.  

We have amended the relevant wording in the methods to clarify this

For the bulk oil MIX and LTN samples, 430 g algal oil, 440 g distilled water combined with 30 g Tween 40 were added to 100g of pre-mix. For the LE sample, 360 g algal oil and 440 g distilled water were added to 200g of the pre-mix. Both samples were hand stirred and placed in the same water bath for 2 hours. Each solution was hand stirred for 30 seconds every hour. A quantity of the combined Tween 40 and lecithin sample, which was not fully emulsified and did not appear to be phase stable was then re-moved from the water bath (bulk oil MIX sample).

(lines 115 to 121).

  1.  The Authors have explained the development of the LTN nanoemulsion in the Methods, but not the development of the LE nanoemulsion.

       Thank you, we have added further detail to fully outline the development of the LE nanoemulsion sample.

For the bulk oil MIX and LTN samples, 430 g algal oil, 440 g distilled water combined with 30 g Tween 40 were added to 100g of pre-mix. For the LE sample, 360 g algal oil and 440 g distilled water were added to 200g of the pre-mix. Both samples were hand stirred and placed in the same water bath for 2 hours. Each solution was hand stirred for 30 seconds every hour. A quantity of the combined Tween 40 and lecithin sample, which was not fully emulsified and did not appear to be phase stable was then re-moved from the water bath (bulk oil MIX sample).

 (lines 115 to 121).

  1. The revised Manuscript still has some typos. For example, the Authors use the term Ë®course emulsionË® (line: 111) instead of Ë®coarse emulsionË®.

       We have removed this and searched through the article for further typos and amended them as needed.

Reviewer 2 Report (Previous Reviewer 3)

Comments and Suggestions for Authors

The paper's novelty may be compromised if the authors don't specifically clarify how this study differs from prior 3PUFA research. Although they attempted to supply all of the data, certain revisions are still necessary.

Abstract: I suggest condensing the abstract and highlighting the primary research finding.
Introduction: Li et al. worked on a 3PuFA nanoemulsion, as the author noted in line 66. What distinguishes their research from yours?

Results and discussion: I recommend the authors to submit the span number and discuss in Table 2

Line 229: Could you please elaborate on the significance and practical use of adjusting your microemulsion's droplet size during the digestion?

Author Response

Response to Reviewers

Dear Reviewers and Editor

We would like to thank you for providing a further opportunity to submit a revised draft of the manuscript ‘Evaluating the stability and digestibility of an omega-3 algal oil nanoemulsion using an in vitro digestion model.’ for publication in Foods.

We wish to thank the reviewers for their valuable feedback and have sought to incorporate their suggestions where possible, thereby improving the quality of our paper. We have made appropriate changes to the manuscript. Please see below, in red, our response to each of the reviewers’ comments accompanied by associated changes to the manuscript and feedback specific changes added to refer to the revised manuscript file with line number.

Reviewer 2

The paper's novelty may be compromised if the authors don't specifically clarify how this study differs from prior 3PUFA research. Although they attempted to supply all of the data, certain revisions are still necessary.

       Thank you for this feedback. We have added more detail to the introduction to clarify how our study is novel and differs from prior LCn-3PUFA research

       The authors have previously established a method to produce nanoemulsions of LCn-3PUFA algal oil with optimized processing conditions and selected emulsifiers (lecithin, Tween 40 and equal ratios of both) to increase the oil load to 50% (w/w) us-ing ultrasonic processing [24]. Furthermore, the authors have also shown that DHA absorption from algal oil nanoemulsions created using lecithin and ultrasound pro-cessing was significantly higher than bulk algal oil, in an in vivo single-blind, ran-domized crossover human trial [30]. In addition, the results of in vitro digestion stud-ies with algal oil and lecithin nanoemulsions prepared with microfluidization demonstrated that the stability of the nanoemulsion had a positive impact on the in-dicator of bioavailability [19]. However, at low pH (pH 1.6) the destabilization of the nanoemulsion resulted in a decrease of DHA hydrolysis compared with the nanoemulsion at pH 4. Studies have yet to investigate combinations of natural and synthetic emulsifiers to stabilize LCn-3PUFA nanoemulsion systems created using ultrasound processing at low pH conditions in digestion.

The current study aimed to conduct a novel evaluation of the stability and in vitro digestibility of LCn-3PUFA algal oil-in-water nanoemulsions created using homoge-nization and ultrasound processing with natural and synthetic emulsifiers in com-parison to an unprocessed hand mixed bulk oil sample containing the same compo-nents.

 (lines 66 to 83).

Abstract: I suggest condensing the abstract and highlighting the primary research finding.

Thank you for this feedback. We have condensed and reordered the abstract to give more focus on the primary research findings

The results showed the creation of nanoemulsions of LCn-3PUFA algal oils offers potentially significant increases in the bioavailability of DHA in the human body. This increase in digesti-bility can be attributed to the smaller particle size of the nanoemulsions, which allows for higher absorption in the digestive system. The LE and LTN nanoemulsions had average droplet sizes of 0.340 ± 0.00 µm and 0.267 ± 0.00 µm respectively, but the bulk algal oil MIX (created with same the components as the LTN nanoemulsion, hand mixed, not processed by rotor-stator ho-mogenizer and ultrasound) had an average droplet size of 73.6 ± 6.98 µm. The LTN algal oil nanoemulsion was stable in the gastric and duodenal phases without detectable destabilization, however the LE nanoemulsion showed signs of oil phase separation in the gastric phase. Under the same conditions, the DHA digested from the LTN nanoemulsion was 47.34 ± 3.14 mg/g, compared to 16.53 ± 0.45 mg/g from the algal oil MIX, showing DHA digestibility from the LTN nanoemulsion was 2.86-fold higher. The findings of this study contribute to the insight of in vitro DHA digestion under different conditions.

(lines 21 to 33).

Introduction: Li et al. worked on a 3PuFA nanoemulsion, as the author noted in line 66. What distinguishes their research from yours?

The Li et al. paper is our work (please see author details in the reference list) in which we investigated the development and integration of LCn-3PUFA nanoemulsions into functional foods, we did not investigate in vitro digestion of the systems in this work, and it has not been investigated in this form in any previously published literature. We have made amendments to the manuscript to clarify the novelty of the current study and justify our methods showing that this was our previous work to create and develop the nanoemulsion systems.

The authors have previously established a method to produce nanoemulsions of LCn-3PUFA algal oil with optimized processing conditions and selected emulsifiers (lecithin, Tween 40 and equal ratios of both) to increase the oil load to 50% (w/w) us-ing ultrasonic processing [24]. Furthermore, the authors have also shown that DHA absorption from algal oil nanoemulsions created using lecithin and ultrasound pro-cessing was significantly higher than bulk algal oil, in an in vivo single-blind, ran-domized crossover human trial [30]. In addition, the results of in vitro digestion stud-ies with algal oil and lecithin nanoemulsions prepared with microfluidization demonstrated that the stability of the nanoemulsion had a positive impact on the in-dicator of bioavailability [19]. However, at low pH (pH 1.6) the destabilization of the nanoemulsion resulted in a decrease of DHA hydrolysis compared with the nanoemulsion at pH 4. Studies have yet to investigate combinations of natural and synthetic emulsifiers to stabilize LCn-3PUFA nanoemulsion systems created using ultrasound processing at low pH conditions in digestion.

 (lines 66 to 79). 

Results and discussion: I recommend the authors to submit the span number and discuss in Table 2

We have now amended Table 2 to include the required data and span number for each of the samples

Table 2. Mean of diameters of droplet size (d32) and span data [36] for algal oil MIX, LE and LTN nanoemulsions before digestion.

Algal oil MIX

LE Nanoemulsion

LTN nanoemulsion

Mean droplet size (d32 (µm))

73.6 ± 6.98

0.340 ± 0.00 a

0.267 ± 0.00 a

Span

1.24

2.91

1.78

                        a Significantly different to the droplet sizes of algal oil MIX (p < 0.05)

(lines 228 to 234).

Line 229: Could you please elaborate on the significance and practical use of adjusting your microemulsion's droplet size during the digestion?

We have added further detail to clarify our findings. The addition of bile extract, phospholipids, pancreatin and pancreatic lipase contributed to the increase in droplet sizes we did not make any intentional adjustments during digestion

The average droplet size of LTN nanoemulsions before digestion was 0.267 ± 0.00 µm and it was measured at 0.245 ± 0.00 µm and 0.238 ± 0.01 µm at 5 min and 60 min in the gastric phase (pH 1.6). There were no significant changes in the droplet sizes, show-ing the LTN sample had good stability in the gastric digestion phase. Contrastingly, the algal oil MIX had an average droplet size of 73.6 ± 6.98 µm before digestion, but it was significantly reduced to 50.1 ± 1.81µm and then 28.83 ± 0.06 µm at 5 min and 60 min digestion respectively (p < 0.05), showing the algal oil in the MIX sample was further emulsified during gastric digestion with shaking. However, the droplet size was still significantly higher (p < 0.05) and over 100-fold larger than the average droplet size of the LTN nanoemulsion. Following gastric digestion, 3.5ml of SBF with phospholipids, bile salts and 7.5ml SDF with pancreatin and pancreatic lipase were added to digestion mixture with SGF, the average droplet size of digestion mixture for the LTN nanoemulsion increased to 0.326 ± 0.01 µm at 5 min, then to 14.20 ± 0.26 at 60 min and further decreased to 8.87 ± 2.56 µm after 180 min of duodenal phase (pH 6.8), showing the mean of droplet size changed significantly (Ñ€ < 0.05). It appears that the addition of bile extract, phospholipids, pancreatin and pancreatic lipase con-tributed to the increase in droplet sizes from 0.326 ± 0.01 µm to 14.20 ± 0.26. Mean-while, the droplet size of the digestion mixture of the algal oil MIX was in larger ranges with a significant (p < 0.05) decrease from 27.30 ± 0.30 µm to 18.97 ± 8.19 µm at the end of the duodenal phase.

(lines 245 to 264).

Round 2

Reviewer 1 Report (Previous Reviewer 2)

Comments and Suggestions for Authors

The resubmitted study entitled “The stability and digestibility of a plant-based omega-3 nanoemulsion using an in vitro digestion model” by Authors Qiqian Zhou, Katie E. Lane and Weili Li, investigated the possibility of usage of lecithin (LE) and a mixture of Tween 40 and lecithin (LTN) to develop nanoemulsions using algal oil as the oil phase. Apart form the physico-chemical properties, the digestibility of the emulsions was also investigated. The resubmitted study has some flaws and unclear statements which should be resolved.

1.  The control in this study is inadequate. Whereas the investigated emulsions, i.e., nanoemulsions, were prepared by mild stirring (by shaking water bath and by hand), as well as rotor-stator and ultrasound homogenization, the control emulsion was obtained only using shaking water bath and hand stirring. The study would be much more beneficial to the readers if  the control was prepared using rotor-stator homogenization as well (without ultrasound processing).

2.     Lines 18-19: “A nanoemulsion of LCn-3PUFA algal oil was developed with 6% w/w emulsifiers: lecithin (LE) or an equal ratio of Tween 40 and lecithin (LTN),…”Please make the statement more understandable for the readers. Please write which concentrations of emulsifiers were used in LTN mixture.

3.   Lines 21-24: “The results showed the creation of nanoemulsions of LCn-3PUFA algal oils offers potentially significant increases in the bioavailability of DHA in the human body. This increase in digestibility can be attributed to the smaller particle size of the nanoemulsions, which allows for higher absorption in the digestive system.” The Authors should reverse the order of these statements to make the text more logical and understandable to consumers, for example: “The increase in digestibility can be attributed to the smaller particle size of the nanoemulsions, which allows for higher absorption in the digestive system. This showed that the creation of nanoemulsions of LCn-3PUFA algal oils offers potentially significantly increase the bioavailability of DHA in the human body.”

4. The Authors often state, throughout the Manuscript, that the nanoemulsions were prepared using “homogenization and ultrasound processing”. However, it is well known that ultrasound processing of a primary emulsion (prepared by rotor-stator homogenizer) also represents homogenization. This is called secondary homogenization [1]. Therefore, the nanoemulsions were prepared by rotor-stator and ultrasound homogenization. The Authors should correct this throughout the Manuscript.

Reference:

1.      McClements DJ. Biopolymers in food emulsions. In: Modern biopolymer science 2009 Jan 1 (pp. 129-166). Academic press.

5.   Line 126: The Authors should state that the “ratio” is actually mass ratio.

6.   The following statement is unclear: (lines: 128-130): “For the bulk oil (360 g)MIX and LTN samples, 430 g algal oil, 440 g distilled water (440 g) was combined with 30 g Tween 40 were added, hand mixed, to 100g of pre-mix.” Please clarify the statement to make it more understandable.

7.  Lines 132-133: “Each solution was hand stirred for 30 seconds every hour”. By mixing oil and water solutions usually emulsion is formed, not solution. The Authors should use some other term instead of “solution” in the cited statement.

8.   The Authors use various terms, i.e., abbreviations, for the emulsion of algal oil prepared by hand stirring, i.e., bulk algal oil MIX, bulk oil MIX, algal oil MIX, algal MIX. The Authors should use only one term, i.e., one abbreviation, for the emulsion throughout the Manuscript.

9. Tables should be self-explanatory. The text in Table 5 (line: 439) explaining what the presented data represent should not be deleted.

10. The title of the Manuscript should be more informative and should include which emulsifiers were used for the preparation of the investigated nanoemulsions.

Author Response

Reviewer 1.

  1. The control in this study is inadequate. Whereas the investigated emulsions, i.e., nanoemulsions, were prepared by mild stirring (by shaking water bath and by hand), as well as rotor-stator and ultrasound homogenization, the control emulsion was obtained only using shaking water bath and hand stirring. The study would be much more beneficial to the readers if  the control was prepared using rotor-stator homogenization as well (without ultrasound processing).

We recognise this as a potential study limitation and have added the following to the discussion to clarify:

The In vitro testing was completed with the algal oil MIX sample compared to the nanoemulsion samples to show how the nanoemusion prepared by using rotor-stator and ultrasound homogenization can potentially improve DHA digestibility compared to bulk oil supplements currently available on the market. However, this study did not evaluate the samples after rotor-stator homogenization, prior to ultrasound pro-cessing, which might be a limitation. (line 500 – 506).

  1. Lines 18-19: “A nanoemulsion of LCn-3PUFA algal oil was developed with 6% w/w emulsifiers: lecithin (LE) or an equal ratio of Tween 40 and lecithin (LTN),…”Please make the statement more understandable for the readers. Please write which concentrations of emulsifiers were used in LTN mixture.

Thank you we have added this detail to the abstract (line 20 – 21).

A nanoemulsion of LCn-3PUFA algal oil was developed with 6% w/w emulsifiers: lecithin (LE) or an equal ratio of Tween 40 (3%) and lecithin (3%) (LTN), 50% w/w, algal oil, and 44% w/w water using a rotor-stator and ultrasound homogenization.

  1. Lines 21-24:“The results showed the creation of nanoemulsions of LCn-3PUFA algal oils offers potentially significant increases in the bioavailability of DHA in the human body. This increase in digestibility can be attributed to the smaller particle size of the nanoemulsions, which allows for higher absorption in the digestive system.” The Authors should reverse the order of these statements to make the text more logical and understandable to consumers, for example: “The increase in digestibility can be attributed to the smaller particle size of the nanoemulsions, which allows for higher absorption in the digestive system. This showed that the creation of nanoemulsions of LCn-3PUFA algal oils offers potentially significantly increase the bioavailability of DHA in the human body.”

We have amended the abstract as you have kindly suggested in your example (line 24 – 28).

The increase in digestibility can be attributed to the smaller particle size of the nanoemulsions, which allows for higher absorption in the digestive system. This showed that the creation of nanoemulsions of LCn-3PUFA algal oils offers a potentially significantly increase the bioavailability of DHA in the human body.

  1. The Authors often state, throughout the Manuscript, that the nanoemulsions were prepared using “homogenization and ultrasound processing”. However, it is well known that ultrasound processing of a primary emulsion (prepared by rotor-stator homogenizer) also represents homogenization. This is called secondary homogenization [1]. Therefore, the nanoemulsions were prepared by rotor-stator and ultrasound homogenization. The Authors should correct this throughout the Manuscript.

Thank you, we have searched and corrected the manuscript accordingly and added the reference you suggest, please see line 89 – 90.

Reference:

  1. McClements DJ. Biopolymers in food emulsions. In: Modern biopolymer science 2009 Jan 1 (pp. 129-166). Academic press

5. Line 126:The Authors should state that the “ratio” is actually mass ratio.

We have amended this on line 116.

  1.  The following statement is unclear: (lines: 128-130): “For the bulk oil (360 g) MIX and LTN samples, 430 g algal oil, 440 g distilled water (440 g) was combined with 30 g Tween 40 were added, hand mixed, to 100g of pre-mix.” Please clarify the statement to make it more understandable.

We have made a slight change to make the method clearer (line 119).  

For the algal oil MIX and LTN samples, 430 g algal oil and 440 g distilled water combined with 30 g Tween 40 were added to 100g of the pre-mix.

  1. Lines 132-133:“Each solution was hand stirred for 30 seconds every hour”. By mixing oil and water solutions usually emulsion is formed, not solution. The Authors should use some other term instead of “solution” in the cited statement.

We have changed this to ‘sample’ (line 123)

Each sample was hand stirred for 30 seconds every hour

  1. The Authors use various terms, i.e., abbreviations, for the emulsion of algal oil prepared by hand stirring, i.e., bulk algal oil MIX, bulk oil MIX, algal oil MIX, algal MIX. The Authors should use only one term, i.e., one abbreviation, for the emulsion throughout the Manuscript.

Thank you, we have searched and corrected all references to the sample to standardise ‘algal oil MIX’,

9. Tables should be self-explanatory. The text in Table 5 (line: 439) explaining what the presented data represent should not be deleted.

We have left table 5 as presented in previous drafts, the text has not been deleted (420 – 425)

  1. The title of the Manuscript should be more informative and should include which emulsifiers were used for the preparation of the investigated nanoemulsions.

We have added the specific emulsifiers to the title to make it more informative.

Reviewer 2 Report (Previous Reviewer 3)

Comments and Suggestions for Authors

The authors have significantly improved the manuscript, which can now be accepted for publication.

Author Response

Dear Reviewer 2,

Thank you to accept the manuscript. 

Kind regards

Weili 

This manuscript is a resubmission of an earlier submission. The following is a list of the peer review reports and author responses from that submission.

Round 1

Reviewer 1 Report

Comments and Suggestions for Authors

This manuscript reports on the use of ultrasound technology to produce stable oil-in-water nanoemulsions of algal oil and to improve the digestibility of DHA using an in vitro digestion model. Though the authors have done some general characterization and comparison with similar studies, in my opinion, the innovation and research depth of this article cannot meet the requirements of Foods. In addition, there are also many other issues that need to be revised.

1. English grammar, tense and spelling should be revised through the whole manuscript.

2. Title

The title should be revised. NOTE:Lack of a predicate verb.

3. Introduction

The first paragraph is not about the benefits and need for algae oil as a replacement for fish oil as a DHA supplement. Please rewrite it.

4. Materials and methods

Line 87-89: Both the expression and the data are confusing and difficult to understand. Please rewrite it.

Line 121-161: Lots of unused subscripts errors. In addition, note that it is necessary to include spaces between numbers and units.

Line 181-184: "Obvious differences" is described, but the data in Table 2 were not analyzed for significance.

Why is it that the horizontal and vertical coordinates of all the graphs are not scaled?

5. Results

Please analyze the data in each table to determine its significance. Although Method 2.2.8 states that the significance analysis was performed using ANOVA, the data in the table are not labelled as such.

6. Discussion and Conclusions

Please rewrite this part. The non-significant subject should be not be considered to express in the research paper.

7. References

A large number of references are too old!

Comments on the Quality of English Language

English grammar, tense and spelling should be revised through the whole manuscript.

Reviewer 2 Report

Comments and Suggestions for Authors

In the study entitled “The stability and digestibility of a plant-based omega-3 nanoemulsion using an in vitro digestion model” by Authors Qiqian Zhou, Katie E. Lane and Weili Li, the stability of nanoemulsions obtained using hand stirring, a homogenizer, and an ultrasonic processor, was investigated. The oil-in-water nanoemulsions were prepared having algal oil as the oil phase and lecithin or a mixture of lecithin and Tween 40 (LTN) as emulsifiers. Furthermore, the digestibility of LTN nanoemulsion was investigated. As a control, an emulsion prepared with hand stirring, and the same emulsifiers, was used. The topic of the study as well as the obtained results are very interesting. However, the Manuscript has many typos which should be corrected. Also, some unclear sentences require clarification. The comments and questions regarding the Manuscript are listed below:

1.   The main concern regarding the Manuscript is the use of the not the most suitable control. Namely, the LTN nanoemulsion was prepared using hand stirring, a homogenizer, and an ultrasonic processor, while the control emulsion (algal oil MIX) was prepared only by hand mixing. Although this is not a deal-breaker, the control should have been prepared using hand stirring and the homogenizer, without using the ultrasonic processor. In this way, the study could show how ultrasonic processing influences the stability and digestibility of the emulsions. Also, the emulsions prepared using only hand stirring are usually not stable and not very suitable as controls in tests performed in the study.

2. Lines 22-24 (Abstract): “The digestibility of stabilized LTN nanoemulsion increased by 200% compared to algal oil MIX (with same emulsifiers of LTN nanoemulsion)…” It is unclear what is algal oil MIX, in the Abstract. The Authors should clarify this.

3.   Lines 59-61:However, at low pH (pH = 1.6) of the gastric phase, the destabilization of nanoemulsion resulted in a reduction of DHA hydrolysis compared with the stable nanoemulsion at gastric phase at pH 4.” Have the Authors meant that at low pH the destabilization of nanoemulsion resulted in an increase of DHA hydrolysis compared with the nanoemulsion at pH 4? The Authors are advised to check reference 24 once again.

4.  Lines 66-72: This section of the Manuscript should be clarified and expanded to convey the aims of the study more precisely. Namely, the Authors claim that they investigated “… the release of omega-3 fatty acids in the algal oil nanoemulsion and algal oil MIX (with same emulsifiers of LTN nanoemulsion )…”. However, the Authors do not explain what are algal oil MIX and LTN nanoemulsion, in the Introduction. Apart from this clarification, the Authors should expand this part of the Manuscript to explain that they have investigated nanoemulsion using lecithin as an emulsifier (LE nanoemulsion) as well.

5.  Lines 74-82 (Materials): The text has many typos, which should be corrected. For example: “L-α-Phosphatidylcholine (P3644-100G) of soybean, Type IV-S. ≥30%”. It should be “from soybean” and there should be comma after Type IV-S, instead of period: “Type IV-S,”. Furthermore, “Polyoxyethylenesorbitan”, “Sulphuric acid”, “Sodium sulphate” should be in lowercase. Please correct this throughout the Manuscript (e.g. line 399). Also correct the following typo: “…was purchased from Fisher Scientific, (UK)”. Delete parentheses or comma.

6.     Lines 87-89: The abbreviation LE, should be explained the first time it was used. After that, only the abbreviation should be used in the text. Also, it is unclear why the Authors write lecithin and algal oil in uppercase.

7.     Lines 93-95: “The coarse emulsion was homogenized for two minutes at a maximum speed (1200 rpm) in a homogenizer (Silverson Machine Ltd, England)…” Please explain in the text of the Manuscript what type of homogenizer was used. Is this a rotor-stator homogenizer?

8.  Lines 97-100: “The algal oil MIX were also prepared with the same amount of combined equal ratio of emulsifiers (TN and LE), algal oil and water by 30 seconds hand premixing.” I find it somewhat unfortunate that the Authors have not used the homogenizer to prepare the control emulsion, i.e., algal oil MIX, instead of using only hand mixing. The emulsions prepared only by hand stirring are usually unstable. The study would be more valid if the control was prepared by a homogenizer.

9.     Lines 104-108: “LTN nanoemulsion or algal oil MIX sample were made up to 5ml meal sample with water to contain 10% wt algal oil and were kept at 37 °C in a water bath for 15min, then 7.5 ml simulated gastric fluids (SGF) with dispersed pepsin, and pepsin was added to achieved a digestion mixture containing 3.2 mg/ml pepsin and 12.6 107 mg/ml pyrogallol (as an antioxidant).” This statement is unclear and should be revised. The Authors should clarify what it means “…were made up to 5ml meal sample with water…”. Also, it is unclear what “…simulated gastric fluids (SGF) with dispersed pepsin, and pepsin was added to achieved a digestion…” means.

10. Lines 123-131: Have the Authors investigated the droplet size of algal oil MIX?

11. Lines 127-128: “For the emulsion samples, an absorption parameter value of 0.001 using a refractive index ratio of 1.488 for algal oil.” Please clarify this sentence.

12. Lines 129-131: “D3, 2 is the volume/surface diameter mean or Sauter mean, it provides a measure of mean diameter specific surface area. D4, 3 is the mean diameter over volume or DeBroukere mean, it provides a measure of droplet specific surface area.” The statement should be clarified. According to this statement, both D4,3 and D3,2 provide a measure of droplet-specific surface area, when D4,3 is the mean diameter of the particle size based on volume-weighted mean results. Furthermore, it is De Brouckere mean diameter and not “DeBroukere”. The Authors should check this.

13. Line 134: please add the rotation speed in rpm as well.

14. Line 138: “under -20 0C for the further analysis”, please correct this to “… -20 °C…”.

15.  Lines 140-145: Please correct the grammar errors, please write “Hexane” in lowercase, vortex in the past tense.

16. Line 150: Please write “Methanol” in lowercase.

17. Liens 155-156: “After cooling to room temperature, 10ml hexane was added and the inverted once.” Please clarify.

18.  Line 167: “…and to 200 ºC at …”. Please correct this.

19. Line 173: Please add IBM SPSS…

20.  Line 175: “nanoemulsion” should be in plural.

21.  Lines 172-175: Please add significance level.

22. Lines 177-192. It would be beneficial to the readers of the study if the appearance of algal oil MIX emulsion during digestion was also shown in Figure 1.

23. Line 178: “The drop size…”. Please correct this.

24. Line 193: Please change “agal oil” to algal oil. Please write “nanoemulsion” in lowercase in both title and Table 2.

25. Table 2. is unclear. The Authors have to explain what is meant by “algal oil” in the Table and how they carried out particle size analysis of algal oil in the Methods section.

26. Line 203. Please add a punctuation mark after number 1, i.e., “Figure1.”.

27. Please write “in vitro” in italics throughout the Manuscript.

28. Line 210. Please explain what is “algal bulk oil MIX”.

29. Please write “Pepsin” in lowercase throughout the Manuscript.

30. Please write “Gastric” and “Duodenal” phases in lowercase throughout the Manuscript.

31.  Lines 252-260. Why didn’t the Authors investigate droplet size distribution for all periods as in the case of LTN nanoemulsion?

32. Line 263: “Droplet size of control…”. Please write “Droplet size distribution of control…”.

33.  Lines 256-259. Please add some references regarding statements about phospholipid micelles and their size.

34.  Lines 271-274: “…meanwhile the peak of micelles with smaller droplet size in the range of 0.03-0.05 μm appeared at 120 min of duodenal phase and subsequently the peak for the micelles with even smaller size in the range of 0.02-0.04 μm appeared at 180 min of duodenal phase,…”. The Authors should clarify this statement, since the peaks for SD120 and SD180 are visible around 0.1 μm, on Figure 4.

35.  Line 287: “…only small peak at 0.08-0.62 μm appeared at the end of digestion..”. There are no peaks observable in Figure 5 in this range.

36.  Line 372: Please correct typos in the title of Table 5.

37. Line 387: “…leading to release fatty acids…”. Please correct to: “…leading to release of fatty acids…”.

38. Lines 420-423: “… demonstrating the stable the nanoemulsion during has a higher digestibility compared the algal oil MIX.”. Please clarify this sentence.

39. Lines 429-432: “Large oil droplets could not be transferred into the blood stream without forming micelles of around 0.1 μm, because every size oil droplet has to form micelles in order to be absorbed, and hence smaller micelles are more readily to be adsorbed [35, 36, 37].” Please clarify this statement.

40. Lines 437-438: “…demonstrating that the LTN nanoemulsion has 200% increased digestibility of DHA compared with algal oil MIX.” Please explain how have you calculated this percent of increased digestibility.

Reviewer 3 Report

Comments and Suggestions for Authors

Abstract: Please incorporate numerical data to demonstrate the importance of the results

Line 32-33; please rephrase it

Line 51-54: Citing literature may seem outdated. Please check and update the references to

ensure that the most recent and relevant studies are cited.

Line 61: There is no information about ultrasound's advantages in preparing nanoemulsions. Please add a literature review and discuss the novelty of your and others' research on nanoemulsions containing algae oil.

Line 71: LTN stands for?

Line 108: How did you lower the pH to 1.6?

Line 131: I recommend add a span index

Line 181: the same comment, add the results of the Span Index and discuss.

Table 4: Add the statistical significance.
